# Prospective observational study and serosurvey of SARS-CoV-2 infection in asymptomatic healthcare workers at a Canadian tertiary care center

Victor H. Ferreira[1]*, Andrzej Chruscinski[1], Vathany Kulasingam[1], Trevor J. Pugh[1,2,3], Tamara Dus[1], Brad Wouters[1], Amit Oza[1], Matthew Ierullo[1], Terrance Ku[1], Beata Majchrzak-Kita[1], Sonika T. Humar[1], Ilona Bahinskaya[1], Natalia Pinzon[1], Jianhua Zhang[1], Lawrence E. Heisler[2], Paul M. Krzyzanowski[2], Bernard Lam[2], Ilinca M. Lungu[2], Dorin Manase[4], Krista M. Pace[4], Pouria Mashouri[4], Michael Brudno[4], Michael Garrels[1], Tony Mazzulli[5], Myron Cybulsky[1], Atul Humar[1‡], Deepali Kumar[1‡]

1 University Health Network, Toronto, Ontario, Canada, 2 Ontario Institute for Cancer Research, Toronto, Ontario, Canada, 3 University of Toronto, Toronto, Ontario, Canada, 4 University Health Network Digital, Toronto, Ontario, Canada, 5 Sinai Health System, Toronto, Ontario, Canada

‡ These authors are joint senior authors on this work.
* victor.ferreira@uhnresearch.ca

**Data Availability Statement:** All relevant data are within the paper and its Supporting Information files.

## Abstract

Health care workers (HCWs) are at higher risk for SARS-CoV-2 infection and may play a role in transmitting the infection to vulnerable patients and members of the community. This is particularly worrisome in the context of asymptomatic infection. We performed a cross-sectional study looking at asymptomatic SARS-CoV-2 infection in HCWs. We screened asymptomatic HCWs for SARS-CoV-2 via PCR. Complementary viral genome sequencing was performed on positive swab specimens. A seroprevalence analysis was also performed using multiple assays. Asymptomatic health care worker cohorts had a combined swab positivity rate of 29/5776 (0.50%, 95%CI 0.32–0.75) relative to a comparative cohort of symptomatic HCWs, where 54/1597 (3.4%) tested positive for SARS-CoV-2 (ratio of symptomatic to asymptomatic 6.8:1). SARS-CoV-2 seroprevalence among 996 asymptomatic HCWs with no prior known exposure to SARS-CoV-2 was 1.4–3.4%, depending on assay. A novel in-house Coronavirus protein microarray showed differing SARS-CoV-2 protein reactivities and helped define likely true positives vs. suspected false positives. Our study demonstrates the utility of routine screening of asymptomatic HCWs, which may help to identify a significant proportion of infections.

## Introduction

SARS-CoV-2 is a novel respiratory coronavirus that has evolved into a widespread global pandemic [1]. The transmission of COVID-19 to healthcare workers (HCWs) from patients, colleagues, or the community is a serious concern as it places potentially highly vulnerable

**Funding:** The study was funded by a peer-reviewed grant from the Mount Sinai Hospital and University Health Network Academic Medical Organization and the Toronto General and Western Hospital Foundation (both DK). This study was conducted with the support of the Genomics (genomics.oicr.on.ca) and Diagnostic Development programs of the Ontario Institute for Cancer Research through funding provided by the Government of Ontario (TP).

**Competing interests:** The authors have declared that no competing interests exist.

patients at risk. HCWs appear to be at higher risk for SARS-CoV-2 infection [2]. Symptom screening for HCWs is a standard infection control practice and mitigates spread to patients and other HCWs. However, studies have shown that a significant proportion of individuals have asymptomatic or pre-symptomatic infection but may still transmit virus [3–7]. The purpose of our current study was to understand the prevalence of asymptomatic SARS-CoV-2 infection in HCWs in a large Canadian tertiary care center in order to determine the potential benefits of asymptomatic HCW screening in hospital settings. This was done by a) screening asymptomatic patients with SARS-CoV-2 PCR and b) performing a serosurvey on a subset of asymptomatic HCWs. As a secondary objective, we sought to validate a novel SARS-CoV-2 protein microarray against commercial serologic assays.

## Materials and methods

### Study overview

This study was approved by the University Health Network's institutional research ethics board; written consent was obtained. The setting for the study was the University Health Network, a large tertiary care center in Toronto, Canada with multiple sites and approximately 1,300 total inpatient beds and 12,000 HCWs. The center includes both acute and long-term facilities, a provincial referral unit for advanced lung support for COVID-19 patients, and several dedicated COVID units. Over a six-week period, HCWs were prospectively enrolled and underwent one to six serial nasopharyngeal (NP) swabs for SARS-CoV-2 PCR testing; communication and action in response to positive results were performed in real-time. HCWs were required to be asymptomatic and not have a previous diagnosis of COVID-19. Symptoms compatible with COVID-19 included fever, headache, new or worsening cough, shortness of breath, sore throat, rhinorrhea, diarrhea, anosmia, myalgias, and conjunctivitis. Additional HCWs, whether asymptomatic or symptomatic, who sought voluntary screening through OHS during the same six-week period, were also included as a separate cohort. During the study period, the hospital cared for 975 COVID-19 patients of which approximately one-third were inpatients. Universal masking was in effect in the hospital and HCWs with direct patient contact were required to wear a face shield. N95 masks were reserved for aerosol generating procedures. During these six weeks, the city of Toronto reported 7,647 new infections in a population of 3 million [8].

### SARS-CoV-2 PCR

NP swabs were collected and underwent PCR testing by the UHN clinical microbiology laboratory using either the Seegene Allplex PCR assay (Seegene, South Korea) or Altona PCR assay (Altona Diagnostics, Germany) using manufacturer's instructions.

### Serology testing

Serologic testing for anti-SARS-CoV-2 IgG antibody was performed on a subset of consenting asymptomatic HCWs with no prior apparent exposure to SARS-CoV-2. Approximately 10mL of peripheral blood was collected (BD Vacutainer, Fisher Scientific, Canada), incubated for at least 30 minutes to allow for clotting, and centrifuged at 2000 RCF for 10 minutes. Serum was collected in cryovials and frozen at -80 for batch processing. Serology was performed using two commercially available IgG assays, one that tests anti-nucleoprotein antibodies by CMIA (Abbott Diagnostics, USA) and the other for anti-spike (S) antibodies (EuroImmun, Germany). Commercial assays were carried out using manufacturer's instructions. The EUROIMMUN anti-SARS-CoV-2 ELISA (IgG) kit [9] (EUROIMMUN AG, Germany) was performed

manually. Briefly, serum was thawed and diluted 1:101 and added to wells pre-coated with antigens corresponding to the S1 region of the spike protein. To detect the bound antibodies, a second incubation was carried out using an enzyme-labelled anti-human IgG and substrate catalyzing a colorimetric reaction. Results are evaluated semi-quantitatively by calculation of the ratio of the extinction of the control or patient sample over the extinction of the calibrator. This ratio was interpreted as follows: < 0.8 negative; ≥ 0.8 to <1.0 borderline; and ≥ 1.1 IgG positive. Reported sensitivity and specificity of this assay is 90% and 100% respectively [10]. The Abbott SARS-CoV-2 IgG [11] assay is a chemiluminescent microparticle immunoassay (CMIA) run on the automated ARCHITECT system (Abbott Laboratories, USA). Briefly, 75uL of undiluted serum per sample was loaded onto SARS-CoV-2 nucleoprotein coated para-magnetic microparticles, and assay diluent were combined and incubated. After washing, an anti-human IgG acridinium-labeled conjugate was added, and the resulting chemiluminescent reaction was measured in relative light units (RLUs). The presence or absence of IgG antibodies to SARS-CoV-2 in the sample was determined by comparing the chemiluminescent RLU in the reaction to the calibrator. An index measurement ≥1.4 was considered positive for anti-SARS-CoV-2 IgG antibodies. The sensitivity and specificity of this assay is 100% and 99.6% respectively [10]. Both antibody tests received Emergency Use Authorization from the US Food and Drug Administration (FDA); the Abbott test has also received Health Canada authorization.

## Protein microarray

To confirm antibody specificities a custom microarray was performed using commercially available Coronavirus recombinant proteins. The Coronavirus antigen microarray was generated using previously published protocols for generation of antigen microarrays to screen for autoantibodies in heart failure and transplantation [12, 13]. Antigens were spotted in triplicate onto two-pad FAST nitrocellulose-coated slides (GVS North America, USA) using a Chipwriter Pro microarrayer (Virtek, Canada) with solid pins (Arrayit, USA). Dried slides were blocked overnight at 4°C in PBS containing 5% FBS and 0.1% Tween. The next day, arrays were incubated with patient serum (diluted 1:100 in blocking buffer) for one hour at 4°C. After washing, the slides were incubated for 45 minutes at 4°C with Cy3-labeled goat anti-human IgG and Alexa Fluor 647-labeled goat anti-human IgM (both Jackson ImmunoResearch, USA). After drying, fluorescent intensities of features were quantified using an Axon 4200A microarray scanner (Molecular Devices, USA) with Genepix 6.1 software (Molecular Devices). Median fluorescent intensity minus local background (MFI-B) was determined at 532nm for Cy3, and 635nm for Alexa Fluor 647. The single averaged MFI-B for each antigen was calculated from the features arrayed in triplicate. A diverse collection of 45 Coronavirus antigens corresponding to SARS-CoV-2, SARS-CoV, MERS-CoV and community coronaviruses (CoV-NL63, -HKU1, -229E and -OC43) (Sino Biological, China; ProSci, USA) were used (S1 Table). Antigens were diluted to 0.25 mg/ml in PBS and stored in aliquots at -80°C until the day of microarray printing.

To validate the results, convalescent sera from known COVID-19+ persons (n = 7) obtained 6 weeks after infection, and sera from healthy controls obtained prior to COVID-19 (n = 18) were tested on the microarray platform. Significance of microarrays (SAM) demonstrated 39 reactivities that were higher in the COVID-19+ sera compared with pre-COVID samples (S3 Fig, S5 Table). The eight highest ranked IgG reactivities by SAM (with mean MFI-B > 1,000 in COVID+ samples) were used for analysis of study samples that were positive by the two commercial kits (anti-NP CMIA and anti-S ELISA). Images of arrays probed with secondary antibodies only, pre-COVID serum, and COVID-19+ serum are shown in S2A Fig.

The linearity of the array assay for antibody detection was demonstrated by probing arrays with serial dilutions of serum from a COVID-19+ person (S2B and S2C Fig).

## Viral genome sequencing

Targeted sequencing of the SARS-CoV-2 genome was performed for NP swab samples that were positive by PCR. Briefly, RNA was isolated from NP swab fluid using Mag–Bind Viral DNA/RNA 96 Kit, and RT-PCR was performed using SuperScript IV First Strand Synthesis System (Thermo Fisher, Canada) and Q5 Hot Start High-Fidelity DNA Polymerase (New England BioLabs Inc., USA). The complete viral genome was amplified using a set of overlapping PCR primers, developed by the ARTIC network [14]. PCR products were sequenced on an Illumina MiSeq system using 250 bp paired-end reads. Reads were aligned to a SARS-CoV-2 reference genome (GenBank: MN908947.3) using a Nextflow workflow [15] that generates a consensus sequence from Illumina reads using the ARTIC network nCoV-2019 novel coronavirus bioinformatics protocol [14]. Consensus calls required a minimum coverage depth of 10, with a frequency threshold of 0.75 to call a variant. Only samples with >75% of the SARS-CoV-2 genome having consensus calls were used.

## Statistical analysis

We hypothesized the asymptomatic disease prevalence of healthcare workers to be 1% and seroprevalence to be 3%. For a cross-sectional study with a type 1 error of 5% and precision of 5%, a sample size of 375 was needed. Due to uncertainties about retention of participants, and to allow us to offer asymptomatic screening to all interested HCWs, we recruited a larger sample size. Analysis was performed using descriptive statistics. Chi-squared test was used to calculate risk factors associated with positivity. All analysis was performed using SPSS version 25 (IBM, USA) and Prism, version 8 (GraphPad, USA). All data are available upon request.

## Results

Three separate cohorts were analyzed (Fig 1). The primary study cohort was a total of 1,669 asymptomatic HCWs that were enrolled over a six-week period (April 17 –May 29, 2020) with a total of 3,173 NP swabs performed. HCWs primarily included nurses (n = 655), physicians (n = 152) and allied health professionals (n = 446), among others (S2 Table). Absence of symptoms was confirmed for all participants at the time of testing. A total of 472/1,555 (29.1%) were actively involved in the care of patients with COVID-19 in the immediate two weeks prior to at least one of their swabs. The second cohort consisted of an additional 4,107 asymptomatic HCWs who were tested voluntarily through OHS. The third cohort consisted of 1,597 HCWs symptomatic HCWs who self-identified as having at least one symptom compatible with COVID-19. The latter two cohorts were added post-hoc to the original study in order to put the study data in context. Cohorts 2 and 3 were diagnosed during the same six-week period as cohort 1.

## SARS-CoV-2 PCR

The prevalence of a positive NP swab at any time point in cohort 1, the primary asymptomatic cohort, was 9/1,669 (0.54%, 95% CI 0.28–1.02). Nurses were more likely positive than other professions (p = 0.003) although taking care of a patient with COVID-19 in the two weeks prior to testing did not increase the likelihood of asymptomatic infection (p = 0.99). Of the nine asymptomatic HCWs who tested positive for SARS-CoV-2, four (44.4%) subsequently developed symptoms while the rest (55.6%) remained asymptomatic (S3 Table). In the

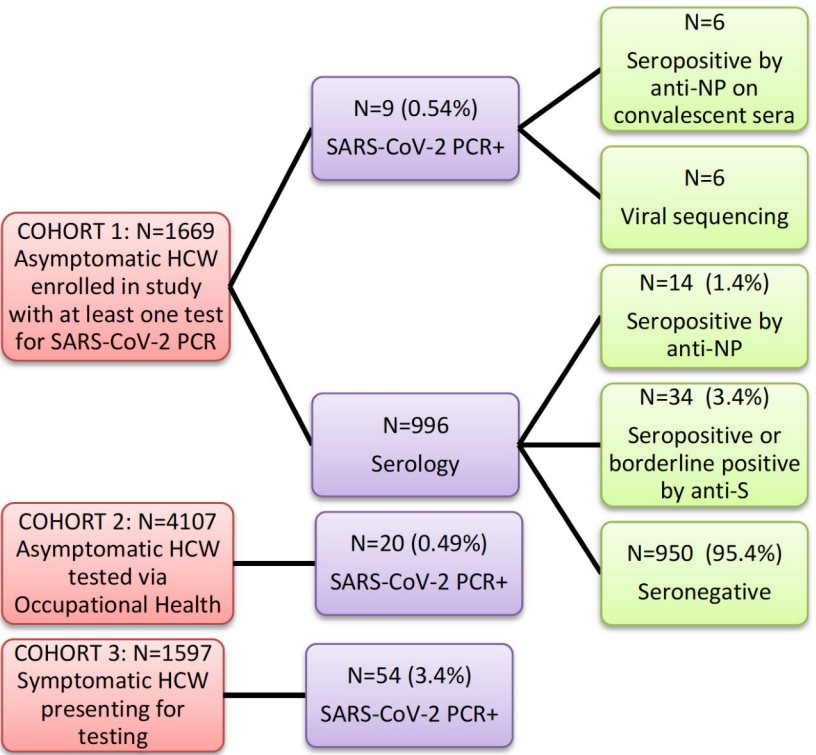

**Fig 1. Study flow and outcomes.** Abbreviations: HCW—healthcare workers, NP—nucleoprotein, S—spike.

secondary cohort of 4,107 asymptomatic HCWs, who presented for voluntary screening at occupational health services, 20 were positive (0.49%, 95%CI 0.32–0.75) for a combined swab positive prevalence of 29/5,776 (0.50%, 95%CI 0.35–0.72) in the two asymptomatic cohorts. In the third cohort, made up of symptomatic HCWs, 54/1597 (3.4%) tested positive for SARS-CoV-2. Based on this, the ratio of symptomatic to asymptomatic positive HCWs during the six-week period was approximately 6.8 to 1.

## Sequencing

We performed whole genome sequencing of SARS-CoV-2 on all positives swab specimens identified in our primary group of asymptomatic HCWs (cohort 1). Sequencing was successful in 6/9 positive HCWs from cohort 1; 3/5 (60%) specimens were from HCWs who remained asymptomatic and on 3/4 (75%) specimens were from participants who developed symptoms. Based on sequence analysis, three predominant viral strains were identified (S1 Fig). In conjunction with an analysis of ward locations for positive HCWs, this suggests that at least some of the positive cases may have been due to HCW-to-HCW transmission or possibly from a common patient source. Interestingly, all six individuals carried the spike protein D614G mutation, caused by an A-to-G nucleotide mutation at position 23,403 in the reference strain [16, 17].

## Serology testing

A subset of 996 asymptomatic HCWs from cohort 1, with no known prior SARS-CoV-2 exposure, also underwent serology testing (S2 Table) to determine seroprevalence. By the anti-nucleoprotein CMIA serology assay, a total of 14/996 (1.4%) were IgG positive (S4 Table). By

the anti-spike assay, a total of 22/996 HCWs were IgG positive (2.2%) and an additional 12/ 996 (1.2%) had borderline positive results. However, only two HCWs were IgG positive by both assays. We then analyzed all 34 seropositives (excluding borderline positives) via an in-house protein microarray to confirm antibodies against specific SARS-CoV-2 antigens (Fig 2A and 2B). In the 14 HCWs positive by anti-nucleoprotein, 13/14 had evidence of IgG antibodies against SARS CoV-2 nucleoprotein and five had evidence of antibodies against other viral proteins including spike protein and the receptor binding domain. Of the 22 positives by anti-spike ELISA, five had evidence of antibodies against at least one SARS-CoV-2 spike protein and only the two that were positive by both assays had evidence of antibodies against nucleoprotein.

We also performed anti-nucleoprotein antibody testing in the 6/9 asymptomatic HCWs from cohort 1 who tested positive for SARS-CoV-2 by PCR. Serum was collected within 2–8 weeks of SARS-CoV-2 diagnosis to ensure seroconversion. Positive antibody test results were found in 4/6 individuals, with scores well above cutoff (>1.4). Negative antibody test results were found in 2/6 individuals (S3 Table). Neither of these participants developed symptoms during the study period.

## Discussion

We provide the first report of asymptomatic HCW screening and seroprevalence of SARS-CoV-2 in Canada. We demonstrate that routine SARS-CoV-2 PCR screening of asymptomatic HCWs in a large tertiary care hospital was valuable to identify and act upon unrecognized SARS-CoV-2 infection. We also found that in the hospital setting, there were significant numbers of asymptomatic infections with the ratio of symptomatic to asymptomatic HCWs being approximately 6.8:1. Among our primary study cohort, we found that nearly half of asymptomatic HCWs who tested positive remained asymptomatic throughout their clinical course, while the other half developed symptoms, a result that is in line with other studies [18]. Serology demonstrated a higher rate of positivity suggesting that additional sequential PCR screening over time would likely be useful.

Previous studies of HCWs have shown mixed results with regards to symptomatic and asymptomatic infections. In Seattle, Washington, 185/3,477 (5.3%) of symptomatic HCWs tested positive for SARS-CoV-2 by PCR [19]. Hunter et al. screened 1,654 HCWs in England and found a 14% rate of positivity with similar rates in non-clinical staff vs. clinical staff [20]. However, no data on symptoms was available in this study. In terms of asymptomatic infection, Lai et al. tested 335 asymptomatic HCWs in Wuhan, China and found 3 positives (0.9%) [7]. No serologic testing was performed in either study. Between March and April of 2020, Fusco et al [21] found that among 115 asymptomatic HCWs tested, only two (1.74%) tested positive for SARS-CoV-2 via PCR, and another two were IgG seropositive for SARS-CoV-2. From 24 March through 7 April 2020, 546 HCWs at Rutgers University were recruited for SARS-CoV-2 screening. In total, 40 (7.3%) HCWs tested positive, of which 27 (67.5%) reported no symptoms when they were tested. Looking after a patient with COVID-19 was not associated with asymptomatic infection and very few seropositives had looked after patients with COVID-19 suggesting that this is not a risk of asymptomatic infection. However, it should be noted that symptomatic HCWs and those with a known previous diagnosis of COVID were excluded from our study. One advantage of asymptomatic HCW screening is detection of outbreaks. Follow-up investigations in the patient wards of positive HCWs from our study helped identify and subsequently contain two separate outbreaks in which previously unidentified patients were also found to be positive.

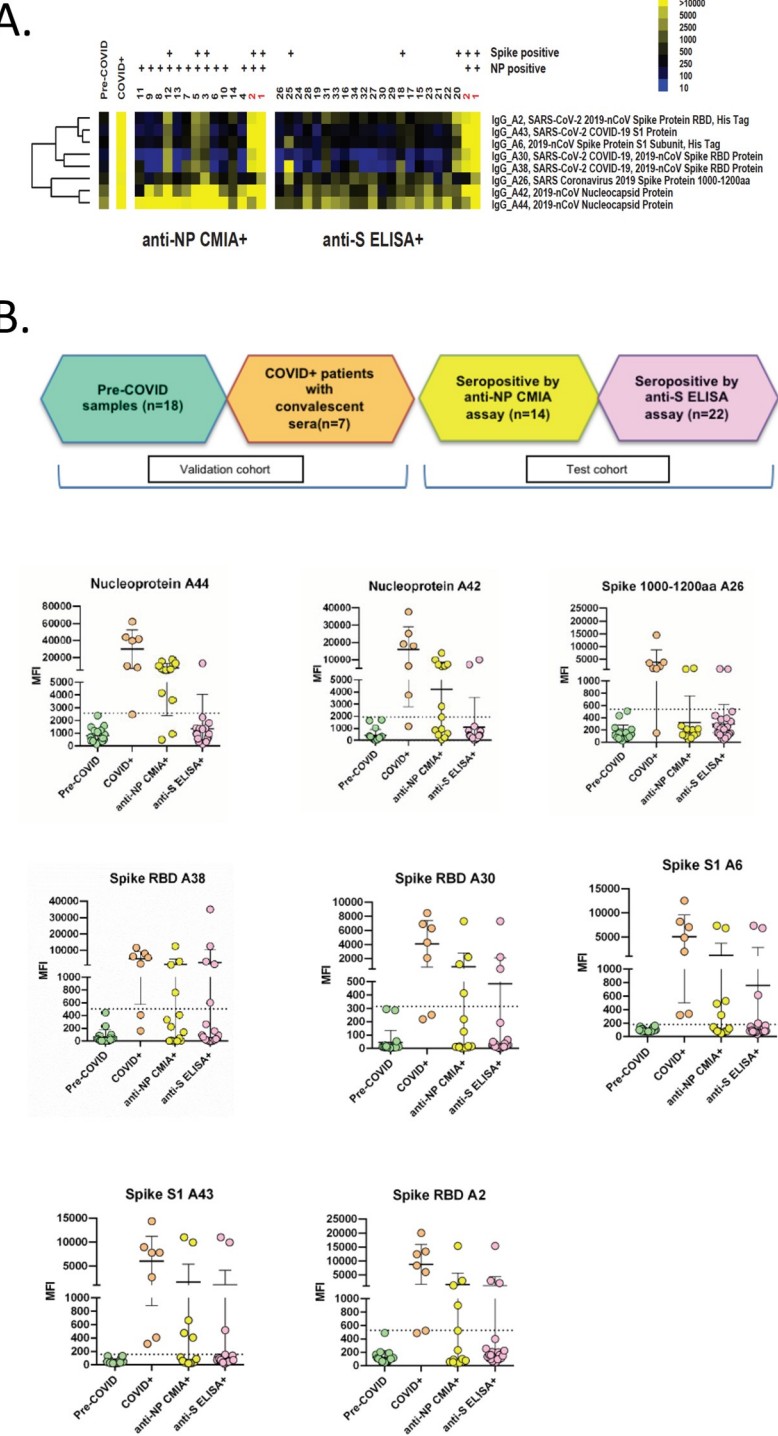

**Fig 2. Heatmap and graph of SARS-CoV-2 reactivity in study samples on antigen microarrays.** A) Heatmap of eight SARS-CoV-2 IgG reactivities in individual anti-NP CMIA+ and anti-S ELISA+ samples. These antigen reactivities represent the eight highest ranked IgG reactivities that are upregulated in COVID+ samples (with mean MFI-B > 1000) as determined by significance analysis of microarrays. Anti-NP and anti-spike reactivity in individual samples on the arrays is indicated above the sample numbers. Positivity on the arrays was determined as described below. The mean reactivity of pre-COVID and COVID+ samples is shown as a comparison. The sample numbers in red indicate dual positive (anti-NP CMIA+ and anti-S ELISA+) samples. Yellow indicates high reactivity, whereas blue indicates low reactivity on the heatmap. B) Graphs of individual antigen IgG reactivity (MFI-B) in pre-COVID, COVID+, anti-NP CMIA+ and anti-S ELISA+ groups. Graphs show mean ± SD for samples in each of the groups.

Samples in the anti-NP CMIA+ and anti-S ELISA+ groups were considered positive if the MFI-B was higher than the mean + 3 SD of the pre-COVID samples (dotted line). Abbreviations: MFI-B—median fluorescent intensity minus background; NP—nucleocapsid protein; S—spike; SD—standard deviation.

In our study, depending on the commercial assay, 1.4%-3.4% of HCW had evidence of past SARS-CoV-2 infection. This is similar to a German cohort where the seroprevalence among 406 clinic staff was found to be 2.7% [22], but lower than a Spanish cohort of HCWs where 9.3% were seropositive [23]. A recent report by Public Health Ontario also showed a 1.4% seroprevalence in the Ontario population although the majority of positives were in persons 60 years or older [24]. Assays vary in their sensitivity and specificity as well as target antigen and have been validated in persons that generally have illness leading to hospitalization. Thus, they may not detect the breadth of antibody responses produced in asymptomatic or milder infections. A novel aspect of our study was confirmatory assessment using a microarray-based assay to determine protein-specific SARS-CoV-2 IgG antibodies. This showed varying protein reactivity in HCWs who were seropositive based on commercially available assays although tended to correspond to results from the anti-NP CMIA. These data coupled with lack of agreement between commercial assays highlight the pitfalls and variability of performing large scale serosurveys in lower prevalence asymptomatic populations. In addition, the antibody profile post-infection may differ in individuals depending on clinical course, and assays that look at multiple antigens simultaneously provide more robust information. While single-target assays may perform relatively well in patients with known COVID, when applied to large seroprevalence studies, performance characteristics appear poorer with significant disagreement between tests. Assays vary in their protein targets and since antibodies may wane over time, distinguishing true from false positives may be difficult. We suspect several of the results on a single assay were likely false positives.

Our study has several limitations. We only performed sequencing and antibody testing on a subset of individuals from cohort 1. These analyses were not possible from cohorts 2 and 3 as these cohorts were added post-hoc and research samples were not collected at the time of their testing. Although we were able to collect convalescent sera from six asymptomatic HCWs who tested positive for SARS-CoV-2, we did not have baseline sera from the majority of those who tested positive by PCR in cohort 1, so it is not possible to discern whether they had antibodies prior to their asymptomatic diagnosis. Furthermore, we were only able to fully sequence viral isolates from six of the nine asymptomatic HCWs in cohort 1. Inability to sequence the remaining three samples may have been due to numerous factors including insufficient sample, sample degradation, or possibly even false positive PCR result. Although we have corroborating sequencing and antibody testing for most asymptomatic HCWs from cohort 1 who tested positive by PCR, we cannot rule out that one or more of these may be false positive.

In summary we show that a significant proportion of HCWs during the pandemic may be asymptomatic/pre-symptomatic and propose that if symptomatic HCW in an institution are being diagnosed with COVID, then asymptomatic HCW testing should also be offered. Data on serosurveys in the asymptomatic HCW population need to be carefully interpreted as performance characteristics of assays may vary. However, the generally higher rate of past infections compared to current infections suggests there is utility in sequential screening of asymptomatic HCW by nasopharyngeal swabs.

## Supporting information

**S1 Fig. Sequencing of SARS-CoV-2 from 6/9 healthcare workers with active infection.** Numbers on y-axis correspond to those in S3 Table. A multiple sequence alignment of all

consensus reads and the MN908947.3 reference was generated, then used to build a phylogenetic tree using augur (https://github.com/nextstrain/augur). Variants were called using scripts developed as part of the nCoV-tools package (https://github.com/jts/ncov-tools). Sites with single base substitutions are shown, with N indicating no coverage at the site. For genome completeness, a cut-off of 75% was used to sequence samples. Three of the nine samples did not meet this cut-off. Genome completeness ranged between 83.7–97.1%. Results demonstrate 3 variants. The D614G mutation, which occurs at nucleotide position 23403 of the reference strain, is indicated with an arrow.
(PDF)

**S2 Fig. Images of antigen microarrays and determination of linearity of the array assay.** A) Images of 2-color arrays probed with secondary antibodies only, pre-COVID serum (negative control) and COVID+ serum (positive control). Antigens were spotted in triplicate; green indicates IgG reactivity, whereas red indicates IgM reactivity. On the array probed only with secondary antibodies, only human IgG and human IgM are detected. On the array probed with pre-COVID serum, reactivity against common community coronavirus antigens is detected. On the array probed with COVID+ serum, there are additional SARS-CoV-2 reactivities detected (boxes). Array features are approximately 500 μm in diameter. B) and C) Linearity studies using serial dilutions of COVID+ serum. Graph B shows MFI-B plotted against serum dilutions, whereas Graph C shows log2 transformed MFI-B. Linear responses are observed over a wide range of serum dilutions using log2 transformed MFI-B. Antibody responses become non-linear as MFI-B approaches saturation levels (MFI-B > 60,000). Abbreviations: MFI-B—median fluorescent intensity minus background.
(PDF)

**S3 Fig. Heatmap of the 39 antigen reactivities upregulated in COVID+ patients as determined by significance analysis of microarrays.** The COVID+ samples (n = 7) form a separate cluster from the pre-COVID samples (n = 18) using a hierarchical clustering algorithm. Yellow indicates high reactivity, whereas blue indicates low reactivity.
(PDF)

**S1 Table. Viral antigens included in protein microarray.**
(DOCX)

**S2 Table. Characteristics of health care workers undergoing nasopharyngeal swab (in cohort 1) and serology testing.**
(DOCX)

**S3 Table. Asymptomatic healthcare workers that had positive SARS-CoV-2 PCR (n = 9) in cohort 1.**
(DOCX)

**S4 Table. Healthcare workers that were SARS-CoV-2 anti-nucleoprotein (NP) IgG positive (n = 14).**
(DOCX)

**S5 Table. List of antigen reactivities upregulated in COVID+ patients as determined by significance analysis of microarrays (fold change > 2, false discovery rate < 1%).**
(DOCX)

## Author Contributions

**Conceptualization:** Brad Wouters, Amit Oza, Myron Cybulsky, Atul Humar, Deepali Kumar.

**Data curation:** Victor H. Ferreira, Andrzej Chruscinski, Vathany Kulasingam, Trevor J. Pugh, Tamara Dus, Matthew Ierullo, Terrance Ku, Beata Majchrzak-Kita, Sonika T. Humar, Ilona Bahinskaya, Natalia Pinzon, Jianhua Zhang, Lawrence E. Heisler, Paul M. Krzyzanowski, Bernard Lam, Ilinca M. Lungu, Dorin Manase, Krista M. Pace, Pouria Mashouri, Michael Brudno, Michael Garrels, Tony Mazzulli, Myron Cybulsky, Deepali Kumar.

**Formal analysis:** Victor H. Ferreira, Andrzej Chruscinski, Vathany Kulasingam, Trevor J. Pugh, Matthew Ierullo, Terrance Ku, Beata Majchrzak-Kita, Sonika T. Humar, Lawrence E. Heisler, Tony Mazzulli, Myron Cybulsky, Atul Humar, Deepali Kumar.

**Funding acquisition:** Deepali Kumar.

**Investigation:** Brad Wouters, Amit Oza, Myron Cybulsky, Atul Humar, Deepali Kumar.

**Methodology:** Victor H. Ferreira, Andrzej Chruscinski, Vathany Kulasingam, Trevor J. Pugh, Tony Mazzulli, Myron Cybulsky, Deepali Kumar.

**Resources:** Andrzej Chruscinski, Trevor J. Pugh, Tony Mazzulli.

**Software:** Pouria Mashouri.

**Supervision:** Victor H. Ferreira, Deepali Kumar.

**Validation:** Andrzej Chruscinski, Vathany Kulasingam.

**Writing – original draft:** Victor H. Ferreira, Andrzej Chruscinski, Atul Humar, Deepali Kumar.

**Writing – review & editing:** Victor H. Ferreira, Andrzej Chruscinski, Vathany Kulasingam, Trevor J. Pugh, Tamara Dus, Brad Wouters, Amit Oza, Matthew Ierullo, Terrance Ku, Beata Majchrzak-Kita, Sonika T. Humar, Ilona Bahinskaya, Natalia Pinzon, Jianhua Zhang, Lawrence E. Heisler, Paul M. Krzyzanowski, Bernard Lam, Ilinca M. Lungu, Dorin Manase, Krista M. Pace, Pouria Mashouri, Michael Brudno, Michael Garrels, Tony Mazzulli, Myron Cybulsky, Atul Humar, Deepali Kumar.

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
