## [Decision Letter · Decision Letter 0]

12 Jan 2021

PONE-D-20-34493

Prospective Observational Study of Screening Asymptomatic Healthcare Workers for SARS-CoV-2 at a Canadian Tertiary Care Center

PLOS ONE

Dear Dr. Ferreira,

Thank you for submitting your manuscript to PLOS ONE. After careful consideration, we feel that it has merit but does not fully meet PLOS ONE’s publication criteria as it currently stands. Therefore, we invite you to submit a revised version of the manuscript that addresses the points raised during the review process.

The Authors are expected to address all the criticisms by all Reviewers. In particular, please clarify the study objectives and consider revising the title to fully reflect the study, define clearly symptoms compatible with COVID-19 (Reviewer #1), discuss study limitations and clarify selection of patients for serological test (Reviewer #2). In additional to the above comments, please address,

Supp Table 2, please align the serological test results with the correct occupation.Please clarify if there were any nosocomial outbreaks in the University Health Network during the study period.

We look forward to receiving your revised manuscript.

Kind regards,

Eric HY Lau, Ph.D.

Academic Editor

PLOS ONE

Journal Requirements:

Additional Editor Comments:

The Authors are expected to address all the criticisms by all Reviewers. In particular, please clarify the study objectives and consider revising the title to fully reflect the study, define clearly symptoms compatible with COVID-19 (Reviewer #1), discuss study limitations and clarify selection of patients for serological test (Reviewer #2). In additional to the above comments, please address,

1. Supp Table 2, please align the serological test results with the correct occupation.

2. Please clarify if there were any nosocomial outbreaks in the University Health Network during the study period.

Reviewers' comments:

Reviewer's Responses to Questions

**Comments to the Author**

1. Is the manuscript technically sound, and do the data support the conclusions?

Reviewer #1: Yes

Reviewer #2: Yes

2. Has the statistical analysis been performed appropriately and rigorously? 

Reviewer #1: Yes

Reviewer #2: Yes

3. Have the authors made all data underlying the findings in their manuscript fully available?

Reviewer #1: Yes

Reviewer #2: No

4. Is the manuscript presented in an intelligible fashion and written in standard English?

Reviewer #1: Yes

Reviewer #2: Yes

5. Review Comments to the Author

Reviewer #1: Dear authors, I read carefully your paper. In general my opinion about your work is positive. It is well-written, methodologically valid, the sample is adequate. Before to take a final decision about the publication I have few major concerns and some minor suggestions.

Major concerns:

• My first doubt concerns the structure and objectives of the article. In practice, it is as if numerous and different papers are included in your work. The main focus is the surveillance of HCWs. Then there are two secondary topics: the genetic sequencing of SARS-CoV-2, and the comparison of the performance of the different serological assays. These two secondary topics are not stated in the objectives, nor in the title, and are just mentioned in the introduction. I suggest to specify better the various topics that will be treated since from the introduction. Also, consider removing the information about genetic sequencing completely. This information is off-topic, and adds nothing to what is already known on the subject, even considering the low number of positive HCWs;

• The second doubt is about the level of novelty and originality of the paper. Many reports about HCWs surveillance have been already published. Please better specify in the discussion which are the elements of novelty in your paper, and what your paper adds to the general knowledge about this topic.

Minor suggestions:

• You refer to asymptomatic HCW both in the title and in the short title. This suggest that the focus of your paper are asymptomatic HCWs only. This is not completely correct, because a relevant quote of symptomatic/pauci-symptomatic HCWs are included too. Please consider to revise the titles;

• Among the objective it is listed “to determine the potential benefits of asymptomatic HCW screening in hospital settings”: this approach is already widely used, and many evidences (and the common sense) already validated it. I suggest removing this sentence from the objectives;

• The sentence “Additional HCWs (asymptomatic or symptomatic) self-identified for voluntary screening through OHS” is unclear. Were these HCWs identified by OHS or by themselves?

• About PPE used by HCWs, you refer that face shield are required for close contacts and N95 masks were used for aerosol-producing procedures. Do you refer to all patients or to suspected/confirmed COVID-19 patients only?

• I suggest the remove the sentence “for an approximate infection rate of 0.25%”. To calculate the population infection rate you should know the real total number of infection, and not only those diagnosed;

• In the Results, the cohort 2 was constituted by many HCWs who voluntary were tested for COVID-19. Was the recruitment period of this cohort the same as for cohort 1? The same 6 weeks period?

• The third cohort is represented by HCWs who have “at least one symptom compatible with COVID-19”. You should define and list these symptoms;

• Among studies in asymptomatic HCWs, consider to cite Fusco FM et al, COVID-19 among healthcare workers in a specialist infectious diseases setting in Naples, Southern Italy: results of a cross-sectional surveillance study. J Hosp Infect. 2020 Aug;105(4):596-600. In this study, as in your present paper, a combined PCR and serology approach has been used;

• Please revise the alignment Supplementary Table 2, in the Occupation box.

Reviewer #2: The Authors report on the experience of a Canadian tertiary care center to investigate asymptomatic COVID19 infections among healthcare workers.

The topic is of high relevance right now and, while other similar reports have been published in literature, I think that this one should be shared as well.

I have some comments:

- limitations: I think the discussion should include better highlight limitations of the study. For example, this is not an universal screening beacuse only 1669 HCW out of 12.000 were enrolled. Is it possible that voluntary recruitment led to bias?

- serology testing: the HCWs undergoing serology testing were from the 1669 people cohort enrolled for NF swabs? Figure 1 seems to suggest so, but it is not clear in the text. If yes, did they all tested negative on the NF swab? If now, was a NF swab performed in case of positive serology? Moreover, was IgM serology performed? If not why? How do you deal with positive IgG testing? We are still not sure of the timing of seroconversion.

- female to male ratio: in the NF swabs cohort 1312 were female (vs. 356 males and 1 not binary) and in the serology cohort 781 were female (vs. 215 males). How do you comment this? Do you have mainly female HCWs in your hospital or females are more likely to undergo voluntary testing?

- It would be nice to see if the positivity rate of the NF swab changes during the weeks accordingly with the general number of reported infection in the city and/or the hospital, as it was described in other reports on asymptomatic SARS-CoV-2 infections.

6. PLOS authors have the option to publish the peer review history of their article (what does this mean?). If published, this will include your full peer review and any attached files.

Reviewer #1: **Yes: **Francesco Maria Fusco

Reviewer #2: **Yes: **Claudia Massarotti

---

## [Author Response · Author response to Decision Letter 0]

27 Jan 2021

PONE-D-20-34493

Prospective Observational Study of Screening Asymptomatic Healthcare Workers for SARS-CoV-2 at a Canadian Tertiary Care Center

PLOS ONE

Dear Academic Editor and Reviewers, 

We thank you for the time you have spent reviewing our manuscript and for your encouraging comments. Below you will find a point-by-point response to all critiques raised by the reviewers. Also attached is a separate file labeled “Revised Manuscript with Track Changes” with the revisions as suggested. 

Journal Requirements:

Response: We have made the necessary revisions to meet the style guidelines. 

Response: We have removed the statement. It was not a core part of the research being presented. 

Additional Editor Comments:

The Authors are expected to address all the criticisms by all Reviewers. In particular, please clarify the study objectives and consider revising the title to fully reflect the study, define clearly symptoms compatible with COVID-19 (Reviewer #1), discuss study limitations and clarify selection of patients for serological test (Reviewer #2). In additional to the above comments, please address,

1. Supp Table 2, please align the serological test results with the correct occupation.

2. Please clarify if there were any nosocomial outbreaks in the University Health Network during the study period.

Response: We thank the Editorial Reviewer for highlighting these issues. We have addressed them all including a) revision to the title, b )inclusion of symptoms compatible with COVID, c) study limitations, d) clarifying selection of patients for serological testing, e) aligning serology with correct occupation, and b) nosocomial outbreaks during the study period. There were two outbreaks that were detected during this time due to participants testing positive in our study. This has been mentioned in the Discussion. We have also responded individually to each point below. 

Review Comments to the Author

Reviewer #1: Dear authors, I read carefully your paper. In general my opinion about your work is positive. It is well-written, methodologically valid, the sample is adequate. Before to take a final decision about the publication I have few major concerns and some minor suggestions.

Major concerns:

• My first doubt concerns the structure and objectives of the article. In practice, it is as if numerous and different papers are included in your work. The main focus is the surveillance of HCWs. Then there are two secondary topics: the genetic sequencing of SARS-CoV-2, and the comparison of the performance of the different serological assays. These two secondary topics are not stated in the objectives, nor in the title, and are just mentioned in the introduction. I suggest to specify better the various topics that will be treated since from the introduction. 

Response: We have clarified the objectives and structure of the study in the Introduction. We also reworded the title to better illustrate the scope of the manuscript. We emphasized the fact that we were performing a serosurvey. We also clarified that the sequencing data was meant to complement the diagnostic PCR and made changes to the manuscript to reflect this. We hope this helps to better weave together the different components of the study. 

Also, consider removing the information about genetic sequencing completely. This information is off-topic, and adds nothing to what is already known on the subject, even considering the low number of positive HCWs;

Response: We agree that the number of HCWs was low with regards to sequencing. However, we would respectfully like to keep this only as supplementary material as it helps to clarify whether PCR results may be falsely positive. For example, Sample 9 (Supp Table 3) had a negative convalescent antibody test result despite having a positive PCR. If we did not have supporting sequencing data, this sample could be misinterpreted as a PCR false positive. Because we have sequencing data for this sample, we can confirm presence of virus and failure to induce antibody responses, or perhaps antibody false-negative. 

• The second doubt is about the level of novelty and originality of the paper. Many reports about HCWs surveillance have been already published. Please better specify in the discussion which are the elements of novelty in your paper, and what your paper adds to the general knowledge about this topic.

Response: We agree that other papers on HCW surveillance have been published. However, one very novel aspect of our study is using a microarray-based assay to determine protein-specific SARS-CoV-2 IgG antibodies. This is a novel adaptation of a platform which has not been described before. This is also the first data from a Canadian cohort. 

Minor suggestions:

• You refer to asymptomatic HCW both in the title and in the short title. This suggest that the focus of your paper are asymptomatic HCWs only. This is not completely correct, because a relevant quote of symptomatic/pauci-symptomatic HCWs are included too. Please consider to revise the titles;

Response: The focus of the study was in asymptomatic HCWs. We also included a cohort of symptomatic HCWs to compare rates of asymptomatic to symptomatic infection during the same window of time, however, the majority of the study, including the serosurvey, is primarily focused on asymptomatic HCWs. While there was reference to pauci-symptomatic individuals, we have reviewed the manuscript and made changes where appropriate to ensure language of the manuscript was precise. 

• Among the objective it is listed “to determine the potential benefits of asymptomatic HCW screening in hospital settings”: this approach is already widely used, and many evidences (and the common sense) already validated it. I suggest removing this sentence from the objectives;

Response: We agree that asymptomatic screening of HCW is used at many centres, but its use is controversial at ours and other hospitals. We agree that other studies have provided some evidence for this. However, to garner evidence for asymptomatic screening was the initial goal of the study (since the study was done prior to start of asymptomatic screening at hospitals) and as such provides the evidence, we need to continue this practice. Therefore, we would respectfully like to keep this objective.

• The sentence “Additional HCWs (asymptomatic or symptomatic) self-identified for voluntary screening through OHS” is unclear. Were these HCWs identified by OHS or by themselves?

Response: These healthcare workers voluntarily appeared for screening at occupational health and safety. During this time, our center offered voluntary screening for any HCW, regardless of symptoms. We have now clarified this detail in the Materials and methods. 

• About PPE used by HCWs, you refer that face shield are required for close contacts and N95 masks were used for aerosol-producing procedures. Do you refer to all patients or to suspected/confirmed COVID-19 patients only?

Response: Face shield and surgical mask were required for all patient contact regardless of COVID positivity. N95 were required for aerosol generating procedures with all patients regardless of COVID positivity. 

• I suggest the remove the sentence “for an approximate infection rate of 0.25%”. To calculate the population infection rate you should know the real total number of infection, and not only those diagnosed;

Response: We thank the reviewer for this and have removed the statement as suggested. 

• In the Results, the cohort 2 was constituted by many HCWs who voluntary were tested for COVID-19. Was the recruitment period of this cohort the same as for cohort 1? The same 6 weeks period?

Response: Yes, all participants were recruited from the same 6-week period of time as cohort 1. We have emphasized this point in the revision. 

• The third cohort is represented by HCWs who have “at least one symptom compatible with COVID-19”. You should define and list these symptoms;

Response: Symptoms compatible with COVID-19 included: fever, headache, new or worsening cough, shortness of breath, sore throat, rhinorrhea, diarrhea, anosmia, myalgias, and conjunctivitis. This information has now been added to the Materials and methods section. 

• Among studies in asymptomatic HCWs, consider to cite Fusco FM et al, COVID-19 among healthcare workers in a specialist infectious diseases setting in Naples, Southern Italy: results of a cross-sectional surveillance study. J Hosp Infect. 2020 Aug;105(4):596-600. In this study, as in your present paper, a combined PCR and serology approach has been used;

Response: Thank you for bringing this study to our attention. We have added a reference to this paper in the discussion. 

• Please revise the alignment Supplementary Table 2, in the Occupation box.

Response: We have properly aligned the table, as requested. Apologies for this oversight. 

Reviewer #2: The Authors report on the experience of a Canadian tertiary care center to investigate asymptomatic COVID19 infections among healthcare workers.

The topic is of high relevance right now and, while other similar reports have been published in literature, I think that this one should be shared as well.

Response: We thank the reviewer for this encouraging remark. 

I have some comments:

- limitations: I think the discussion should include better highlight limitations of the study. For example, this is not an universal screening beacuse only 1669 HCW out of 12.000 were enrolled. Is it possible that voluntary recruitment led to bias?

Response: The reviewer is correct in pointing out that this was not universal screening. It was a prospective study where HCW were required to consent to participate. It was near the start of the pandemic in Canada where asymptomatic screening was not standard of care. Therefore, through a study, we sought to determine the utility of asymptomatic screening. The participants were therefore those that volunteered to participate and may be different than those that did not participate. However, we sampled a significant proportion of our HCWs (14%) so believe the risk of bias is low. Nevertheless, we have added this point to the limitations section in the Discussion. 

- serology testing: the HCWs undergoing serology testing were from the 1669 people cohort enrolled for NF swabs? Figure 1 seems to suggest so, but it is not clear in the text. If yes, did they all tested negative on the NF swab? If now, was a NF swab performed in case of positive serology? Moreover, was IgM serology performed? If not why? How do you deal with positive IgG testing? We are still not sure of the timing of seroconversion.

Response: We have made sure to clarify that the serosurvey was performed exclusively on a subset of consenting HCWs from cohort 1 (ie those enrolled for NF swabs). Blood collection for antibody testing and swab collection were performed at the same time. Only one asymptomatic HCW who tested positive for SARS-CoV-2 had serology drawn the same day of their positive PCR. Initial serology was negative but upon follow-up antibody testing several weeks later, this HCW became seropositive. The remainder of nasopharyngeal swabs were negative. IgM serology was not performed. At the time of the study, the majority of validated commercial antibody tests were for IgG antibodies. Neutralizing antibodies are also of the IgG isotype so this information may be of more clinically relevant than IgM. The IgM response also appears to decline much faster (Dan et al., 2021; Science). As such, we focussed primarily on IgG responses. Those with positive IgG testing, and negative contemporaneous PCR, are discussed in Supp Table 4. These HCWs were felt to have past resolved infection since all had negative nasopharyngeal swabs as well. 

- female to male ratio: in the NF swabs cohort 1312 were female (vs. 356 males and 1 not binary) and in the serology cohort 781 were female (vs. 215 males). How do you comment this? Do you have mainly female HCWs in your hospital or females are more likely to undergo voluntary testing?

Response: Yes, at UHN 73% of HCWs are female (https://www.uhn.ca/corporate/AboutUHN/Pages/uhn_at_a_glance.aspx#research). We believe this explains the significant discrepancies in sex. 

- It would be nice to see if the positivity rate of the NF swab changes during the weeks accordingly with the general number of reported infection in the city and/or the hospital, as it was described in other reports on asymptomatic SARS-CoV-2 infections.

Response: We agree that this would be of interest. The number of swabs over the 6 weeks were overall evenly distributed over the six weeks of the study. However, due to the overall small number of positives among asymptomatic HCWs, we do not think a formal analysis is possible. Greater numbers of cases would likely be required to appropriately make this valuable comparison.

---

## [Decision Letter · Decision Letter 1]

4 Feb 2021

Prospective observational study and serosurvey of SARS-CoV-2 infection in asymptomatic healthcare workers at a Canadian tertiary care center

PONE-D-20-34493R1

Dear Dr. Ferreira,

We’re pleased to inform you that your manuscript has been judged scientifically suitable for publication and will be formally accepted for publication once it meets all outstanding technical requirements.

Kind regards,

Eric HY Lau, Ph.D.

Academic Editor

PLOS ONE

Additional Editor Comments (optional):

Reviewers' comments:

Reviewer's Responses to Questions

**Comments to the Author**

1. If the authors have adequately addressed your comments raised in a previous round of review and you feel that this manuscript is now acceptable for publication, you may indicate that here to bypass the “Comments to the Author” section, enter your conflict of interest statement in the “Confidential to Editor” section, and submit your "Accept" recommendation.

Reviewer #1: All comments have been addressed

Reviewer #2: All comments have been addressed

2. Is the manuscript technically sound, and do the data support the conclusions?

Reviewer #1: Yes

Reviewer #2: Yes

3. Has the statistical analysis been performed appropriately and rigorously? 

Reviewer #1: Yes

Reviewer #2: Yes

4. Have the authors made all data underlying the findings in their manuscript fully available?

Reviewer #1: Yes

Reviewer #2: Yes

5. Is the manuscript presented in an intelligible fashion and written in standard English?

Reviewer #1: Yes

Reviewer #2: Yes

6. Review Comments to the Author

Reviewer #1: (No Response)

Reviewer #2: I am satisfied with the answers to my previous remarks and I therefore recommend the article for publication.

7. PLOS authors have the option to publish the peer review history of their article (what does this mean?). If published, this will include your full peer review and any attached files.

Reviewer #1: No

Reviewer #2: **Yes: **Claudia Massarotti

---

## [Editor Report · Acceptance letter]

5 Feb 2021

PONE-D-20-34493R1 

Prospective observational study and serosurvey of SARS-CoV-2 infection in asymptomatic healthcare workers at a Canadian tertiary care center 

Dear Dr. Ferreira:

I'm pleased to inform you that your manuscript has been deemed suitable for publication in PLOS ONE. Congratulations! Your manuscript is now with our production department. 

Kind regards, 

on behalf of

Dr. Eric HY Lau 

Academic Editor

PLOS ONE